# The Effects of High CO_2_ and Strigolactones on Shoot Branching and Aphid–Plant Compatibility Control in Pea

**DOI:** 10.3390/ijms232012160

**Published:** 2022-10-12

**Authors:** Hendrik Willem Swiegers, Barbara Karpinska, Yan Hu, Ian C. Dodd, Anna-Maria Botha, Christine H. Foyer

**Affiliations:** 1School of Biosciences, College of Life and Environmental Sciences, University of Birmingham, Edgbaston, Birmingham B15 2TT, UK; 2Department of Genetics, Stellenbosch University, Stellenbosch 7600, South Africa; 3Centre for Plant Sciences, Faculty of Biological Sciences, University of Leeds, Leeds LS2 9JT, UK; 4Zhejiang Provincial Key Laboratory of Agricultural Resources and Environment, College of Environmental & Resource Science, Zhejiang University, Hangzhou 310058, China; 5Lancaster Environment Centre, Lancaster University, LEC Building, Lancaster LA1 4YQ, UK

**Keywords:** aphids, climate change, high atmospheric carbon dioxide, phytohormones, plant architecture, strigolactones

## Abstract

Elevated atmospheric CO_2_ concentrations (eCO_2_) regulate plant architecture and susceptibility to insects. We explored the mechanisms underpinning these responses in wild type (WT) peas and mutants defective in either strigolactone (SL) synthesis or signaling. All genotypes had increased shoot height and branching, dry weights and carbohydrate levels under eCO_2_, demonstrating that SLs are not required for shoot acclimation to eCO_2_. Since shoot levels of jasmonic acid (JA) and salicylic acid (SA) tended to be lower in SL signaling mutants than the WT under ambient conditions, we compared pea aphid performance on these lines under both CO_2_ conditions. Aphid fecundity was increased in the SL mutants compared to the WT under both ambient and eCO_2_ conditions. Aphid infestation significantly decreased levels of JA, isopentenyladenine, *trans*-zeatin and gibberellin A4 and increased ethylene precursor ACC, gibberellin A1, gibberellic acid (GA_3_) and SA accumulation in all lines. However, GA_3_ levels were increased less in the SL signaling mutants than the WT. These studies provide new insights into phytohormone responses in this specific aphid/host interaction and suggest that SLs and gibberellins are part of the network of phytohormones that participate in host susceptibility.

## 1. Introduction

Atmospheric carbon dioxide concentrations [CO_2_] are already 50% higher than before the industrial revolution and they are predicted to double again over the next 60 years [1]. Elevated atmospheric CO_2_ concentrations (eCO_2_) suppress photorespiration and increase photosynthetic carbon assimilation in C3 plants leading to substantial increases in carbon gain and crop productivity [2]. High CO_2_ also reduces stomatal apertures via the stomatal CO_2_ signal transduction pathways [3]. The CO_2_-dependent regulation of stomatal conductance restricts plant transpiration and enhances water use efficiency. As a result, the Earth has become greener over the last two decades as plants have produced more leaf area [4]. While these changes can be largely attributed to the regulation of photosynthesis, eCO_2_ has a strong impact on the physiology of C3 plants that extends far beyond photosynthesis and C metabolism [5]. For example, plants grown under eCO_2_ have lower concentrations of most mineral nutrients and this effect may diminish crop quality and nutrient cycling in terrestrial agro-ecosystems [6].

Atmospheric CO_2_ levels also exert a strong influence on the susceptibility of plants to pathogens and herbivores [2]. Insect herbivores are predicted to have an increasingly negative impact on crop production in the coming decades, particularly in African countries [7], and this will have serious implications for food security (Martinelli et al., 2015) [8]. However, current concepts of how eCO_2_ will influence plant susceptibility to phloem-feeding insects, such as aphids, remains controversial [9], not least because the literature data are variable [10,11]. Populations of the soybean aphid (*Aphis glycines*) were larger under eCO_2_, a finding that was related to a higher leaf temperature because of less open stomata [2]. While eCO_2_ had little impact on aphid performance in oilseed rape [12], pea aphids (*Acyrthosiphon pisum*) performed better under eCO_2_ on *Medicago sativa* [11], but worse on *Vicia faba* [13].

Aphids are important plant pests that often have a broad host range that significantly reduce the yields of susceptible plants via nutrient depletion, feeding damage to host tissues and the spread of viruses [14]. Aphids are phloem feeders with a probing stylet that wounds plant tissue as the insect searches for the phloem. The stylet secretes saliva that contains elicitors and proteins that modulate plant defensive responses [15]. Aphid feeding triggers the host multi-level immune system that operates to mitigate the adverse effects of pathogens and insects [16]. The host defensive networks that underpin plant responses to aphids have been characterized in a number of plant species [17,18,19,20]. Aphid feeding results in extensive changes to the leaf transcriptome and metabolome signatures [21], revealing a complex interplay between the different hormones regulating plant basal immunity. Changes in the levels of phytohormones such as jasmonic acid (JA), salicylic acid (SA), abscisic acid (ABA) and indole acetic acid (IAA) have been documented [21,22]. However, little attention has been paid to the role of strigolactones (SL) in plant–aphid interactions. SLs fulfil many important roles in the control of plant growth and architecture, seed dormancy and senescence as well as abiotic stress tolerance [23,24,25]. These carotenoid-derived phytohormones are also critical regulators of plant–microbe interactions in the rhizosphere, such as the symbiosis with arbuscular mycorrhizal fungi [26]. They also serve functions in plant responses to biotic stresses as a result of bacterial and fungal pathogens [27,28,29]. Growth under eCO_2_ increased the resistance of two *M. truncatula* genotypes to pea aphids by increasing salicylic acid (SA)-dependent defenses and decreasing jasmonic acid (JA) and ethylene-dependent signaling pathways, as well as increasing the density of non-glandular and glandular trichomes [10].

The pea aphid, *Acyrthosiphon pisum*, can only colonize plant species in the family *Fabaceae*, but can manipulate host defense signaling networks to increase infestation. Better performance of *A pisum* on their native hosts was not attributed to variation in ABA levels (which were down-regulated in all aphid clone-plant combinations), but instead modulation of the SA- and JA-defense signaling pathways [30]. However, the importance of SLs in aphid–plant compatibility or in shoot responses to eCO_2_ have not been characterized. We, therefore, compared shoot architecture and the fecundity of *A. pisum* in wild type peas and different pea *ramosus* mutants (*rms*) [31] under ambient and eCO_2_ conditions. Specifically, we examined responses to eCO_2_ and aphids in mutants that are deficient in either SL synthesis (*rms1-2* and *rms5-3*) or SL signaling (*rms3-1* and *rms4-1*). Since SL-deficient mutants have lower levels of the defense-related hormones JA and SA [29], we hypothesized that aphid performance would be enhanced on these genotypes.

## 2. Results

Wild type plants, the SL synthesis mutants, *rms1-2* and *rms5-3,* and the SL signaling mutants, *rms3-1* and *rms4-1,* were grown in air or eCO_2_ for up to 32 days after germination. The wild type shoots were significantly taller grown under eCO_2_ than those grown in air after 14 and 32 days of growth (Figure 1). While eCO_2_ tended to increase stem height in the SL mutant lines, the effect was only significant in *rms3-1* (Figure 1; Day 32).

The SL-defective mutants were significantly more branched than the wild type from day 14 after sowing onwards (Figure 2). Growth under eCO_2_ significantly increased the branching of the wild type shoots at day 32 (Figure 2). While eCO_2_ tended to increase shoot branching in all the mutant lines, the effect of eCO_2_ was only significant in *rms1-2* (Figure 2; Day 32).

Growth under eCO_2_ significantly decreased the fresh weight/dry weight ratios of the wild type shoots (Figure 3). This effect was absent from the SL-defective mutants, which had significantly higher fresh weight/dry weight ratios than the wild type 28 days after sowing under both growth conditions (Figure 3A). Growth under eCO_2_ significantly increased shoot dry weight of all lines (Figure 3B).

All the lines had similar levels of leaf glucose, fructose, sucrose and starch when plants were grown in air (Figure 4). Growth under eCO_2_ significantly increased the levels of sucrose and starch in all the lines (Figure 4). While eCO_2_ also tended to increase the levels of leaf hexoses (glucose and fructose), this effect was only significant in some lines (Figure 4).

Aphid fecundity was increased in the SL mutants compared to the wild type plants grown under ambient CO_2_ conditions (Figure 5). However, growth under eCO_2_ had no effect on aphid fecundity (Appendix A).

Phytohormone levels were determined in the shoots of plants grown under ambient or eCO_2_ conditions. Since no significant effects of eCO_2_ on the levels of measured phytohormones were detected, we compared phytohormone levels in the absence and presence of aphids under eCO_2_ conditions (Figure 6). Without aphid exposure, the SL mutants had lower levels of JA, SA and gibberellic acid (GA_3_) than the wild type under eCO_2_ conditions.

Aphid infestation changed shoot concentrations of different phytohormones (Figure 6). The presence of aphids had no effect on SA accumulation in the shoots, but the levels of JA, isopentenyladenine, *trans*-zeatin and gibberellin GA_4_ were significantly decreased (Figure 6). In contrast, the levels of the ethylene precursor, 1-aminocyclopropane-1-carboxylic acid (ACC) and GA_1_ were increased after aphid infestation in all lines. Aphid exposure had no statistically significant effect on GA_3_ levels and there were no significant differences in GA_3_ accumulation between WT non-exposed and WT aphid-exposed plants. However, there was differential GA_3_ accumulation between the genotypes as a result of aphid exposure, which explained 18% of the variability in fecundity (Figure 7).

## 3. Discussion

Understanding the role of plant immunity in host susceptibility to aphids is essential to generate durable and sustainable aphid control strategies. The infestation of the pea shoots by the specialist pea aphid not only substantially decreased levels of the key defense hormone JA, but also isopentenyladenine and *trans*-zeatin, which are components of cytokinin (CK) metabolism. CK signaling pathways play a role in plant resistance by regulating SA-dependent defenses, affecting lignification and inducing protective proteins and phytoalexins [32,33]. Taken together, the findings presented here demonstrate that the pea aphids interact with host plants at the molecular level and successfully suppress plant defenses. In these studies, we tested the hypothesis that the performance of the pea aphid, *A. pisum*, would be enhanced on pea genotypes that were defective in SL signaling. The data presented here show that SL mutants are more susceptible to aphid infestation than the wild type plants. Moreover, the results presented here provide new information concerning how the pea aphid manipulates host defense networks to increase infestation. Aphid infestation was accompanied by large decreases in the levels of JA, isopentenyladenine, *trans*-zeatin and gibberellin A4 (Figure 6). While modulation of the SA- and JA-defense signaling pathways has previously been reported in this compatible aphid/host interaction [20], effects on gibberellin metabolism and signaling have not been reported previously.

The levels of JA and SA were decreased in the SL-deficient mutants relative to the wild type, suggesting that lack of SL synthesis and signaling impairs defense hormone accumulation. Although lower JA contents have previously been reported in the leaves of the *Atmax* mutants [34], SLs antagonized the JA pathway in rice roots [35]. The rice SL mutants were also less susceptible to *Meloidogyne graminicola* infection than the wild type plants [35] In addition, systemic-acquired resistance was increased in Arabidopsis SL mutants, and SA levels were lower in the *max2* mutants than wild type [36]. The Arabidopsis SL mutants were more sensitive to infection by the biotrophic pathogen *Pseudomonas syringae* DC3000 [37]. SL-dependent modulation of *At*ACC4** expression has previously been reported [38], but no significant changes in ACC levels were observed in the pea SL mutants compared to the wild type. Taken together with the literature evidences, the data presented here demonstrate that SLs interact with defense hormones to regulate immunity and defenses against phloem-feeding insects. The levels of GA_3_ were lower in the SL mutants than the wild type, significantly so in the *rms4* mutant. This finding may be related to the role of gibberellin signaling in the regulation of SL synthesis [39] (Ito et al., 2017). However, decreased GA_3_ levels have previously been reported in response to aphids [40]. Together, these data suggest that GA_3_ may thus be important in the defenses of pea plants against aphid infestation (Figure 7).

It is interesting that the presence of aphids also led to a large decrease in the levels of GA_4_, but increased GA_3_ and GA_1_ levels. The pea aphids, therefore, also regulate the synthesis and metabolism of these endogenous plant growth regulators. The finding that gibberellin levels were altered in the SL mutants may be significant in enhancing the ability of the aphids to infest the plants. The *rms3-1* mutant is defective in the pea orthologue of the rice D14 SL receptor, which interacts with the GA signaling repressor, SLR1 [41]. Interactions between GA- and SL-mediated control of defense responses are, therefore, likely to be important in the ability of the aphids to infest pea plants. We are currently investigating whether SLs are important in the susceptibility of pea plants to infestation by the generalist (polyphagous) feeder *Myzus persicae* (green peach aphid), which can infest over 400 plant species. Since *M. persicae* can infest most dicotyledonous plants, it will be interesting to compare whether the generalist (polyphagous) feeder is also able to decrease the levels of phytohormones involved in pea growth and defense.

SLs regulate growth redistribution and repress shoot branching by rapidly modulating auxin transport [42]. The highly branched SL-deficient (*rms1* and *rms5*) and SL signaling (*rms3* and *rms4*) mutants are indicative of negative feedback control in which auxin up-regulates SL synthesis in an RMS2-dependent manner and SL down-regulates auxin synthesis in an RMS3- and RMS4-dependent manner [43]. The data presented in this study show that growth under eCO_2_ increased the branching of pea shoots in all genotypes, suggesting that SLs are not required for shoot responses to eCO_2_. The absence of a marked SL-dependent effect may be due to the large increase in soluble sugars that was found in all plants grown under eCO_2_ because sugars are known to suppress the auxin-induced SL pathway that promotes bud outgrowth [44]. However, the fresh weight/dry weight ratios were decreased by eCO_2_ in the wild type, but not in the shoots of the mutants, suggesting that SLs might participate in regulating shoot water content by modulating stomatal conductance. Although SL-deficient tomato and Arabidopsis plants are drought-sensitive because of their constitutively higher stomatal conductance [45], the *rms* pea mutants maintained similar stomatal conductance to WT plants under optimal conditions [46,47]. While further studies of stomatal response to eCO_2_ are necessary to understand these genotypic differences in shoot water status, we can conclude that SLs are not involved in growth regulation under eCO_2_.

In summary, while eCO_2_ has been predicted to have a major impact on plant susceptibility to herbivores, the data presented here shows that aphid–plant compatibility control of phytohormones is not changed by eCO_2._ However_,_ we show that strigolactones and gibberellins are part of the network of phytohormones that regulate the susceptibility of peas to infestation by the pea aphid.

## 4. Materials and Methods

### 4.1. Biological Resources

Wild type pea (L107, Torsdag cultivar) seeds and mutants deficient in either SL synthesis mutants (*rms1-2* and *rms5-3*) or SL signaling (*rms3-1* and *rms4-1*) were provided by Professor Christine Beveridge. Stocks of the aphid *Acyrthosiphon pisum* were maintained on wild type pea at a 16-hour day length and 22 ± 3 °C.

### 4.2. Growth Conditions and Phenotypic Analysis

Seeds of the wild type, *rms3-1*, *rms4-1*, *rms1-2* and *rms5-3* pea mutants were allowed to germinate in vermiculite for 5 days. Thereafter, six seedlings of each genotype were transferred to Levington F2 compost (Evergreen Garden Care Ltd., Camberley, UK) in pots with a top diameter of 14 cm, 9.4-centimeter bottom diameter and a height of 12 cm equating to a pot volume of 1.3 L. These plants were used for experiments 28 days or less in duration. When plants were grown for 32 days, seven seedlings were planted in pots with an 18-centimeter top diameter, 13-centimeter bottom diameter, a height of 16.5 cm and volume of 3.14 L. Plants were grown in Microclima (Snijders Labs, Tilburg, Netherlands) controlled environment chambers at 21 °C, 14 h/10-day/night regime, 70% humidity and either at ambient CO_2_ (420 ± 20 ppm CO_2_) or eCO_2_ (750 ± 50 ppm CO_2_). Plant height and branching were determined at days 7, 14 and 32 after transplanting. Height was determined by measuring the stem from the soil to the apical shoot. Total branching was calculated as the sum of the height and length of each branch, not including branches shorter than 5 mm. Fresh and dry plant weights were determined at 28 days after transplanting, while dry weight was determined at 32 days.

### 4.3. Leaf Carbohydrate Measurements

The youngest mature leaves were harvested from 28-day-old plants grown in ambient and eCO_2_ for 28 days. Leaf carbohydrates were determined spectrophotometrically, as described by Lunn and Hatch (1995) [48]. Briefly, sugars were extracted from 0.1 g of leaf tissue with hot 80% ethanol and centrifuged. Reducing sugars (glucose, fructose and sucrose) were recovered from supernatant and starch was extracted from the pellet. Soluble sugars were measured by the increase in absorbance at *340 nm* as NADPH was produced in coupled enzymatic assays using glucose-6-phosphate dehydrogenase, hexokinase, phosphoglucose isomerase and invertase.

### 4.4. Aphid Fecundity

Wild type (WT), *rms3-1* and *rms4-1* pea mutants were germinated and transplanted in the same media and growth conditions as described above. In order to obtain *A. pisum* nymphs of the same age, adult *A. pisum* were placed on WT pea for 24 h and allowed to produce nymphs. One of these nymphs (foundress) was then placed on a 5-day-old plant from each of the genotypes, directly after transplantation. The aphids were contained on the pea plants using 07ZL flowerpot cages (Insectopia, Austrey, United Kingdom) and allowed to reproduce for 15 days. Ten replicates were performed for each pea genotype at either ambient or elevated CO_2_. The number of aphids present on each plant was then determined.

### 4.5. Phytohormone Levels

Entire plant shoots (comprising leaves, stipules and stems) that had been infected with *A. pisum* were harvested. Metabolism was immediately stopped by immersion in liquid nitrogen. Samples were stored at −80 °C before freeze drying. Five replicates per line (wild type, *rms3-1* and *rms4-1* mutant pea) were analysed for cytokinins (*trans*-zeatin, tZ, zeatin riboside, ZR, and isopentenyl adenine, iP), gibberellins (GA_1_, GA_3_{gibberellic acid} and GA_4_), indole-3-acetic acid (IAA), abscisic acid (ABA), salicylic acid (SA), jasmonic acid (JA) and the ethylene precursor 1-aminocyclopropane-1-carboxylic aid (ACC), according to Albacete et al., (2008) [49] with some modifications, as described by Castro-Valdecantos et al., (2021) [50].

### 4.6. Statistical Analysis

All datasets to be analysed were firstly subjected to a D’Agostino–Pearson normality test as well as Spearman’s test to determine if variances between variables were heteroscedastic (Appendix A). A Grubbs test was used to test for outliers within the aphid fecundity dataset. Thereafter, a two-way ANOVA was performed on datasets conforming to a Gaussian distribution and which were homoscedastic (Appendix A). If an independent variable ([CO_2_], genotype or aphid exposure) had a significant effect on the response (*p* < 0.05), Tukey’s multiple comparison test was used to compare means between genotypes (Appendix A), while Sidak’s multiple comparison test was used to compare [CO_2_] (Appendix A) or aphid exposure (Appendix A) within genotype. Datasets which were normally distributed, but heteroscedastic, were analysed with a Welch ANOVA (Appendix A) test followed by Dunnett’s T3 multiple comparisons test to compare genotype (Appendix A) means or Holm–Sidak’s multiple comparisons test to compare [CO_2_] (Appendix A) or aphid exposure (Appendix A) within genotype. Log transformations were used [Y = Log(Y)] if the transformation resulted in a normal distribution of the non-normally distributed datasets (Appendix A). When transformation did not work, a Kruskal–Wallis test was used to determine if genotype groups had a significant effect on the response measured, while in the case of [CO_2_] or aphid exposure, a Mann–Whitney test was used (Appendix A). Following a significant *p*-value computed by the Kruskal–Wallis test, multiple comparisons were performed using Dunn’s procedure (Appendix A), while a significant effect determined by the Mann–Whitney test was followed by a Holm–Sidak’s multiple comparisons test (Appendix A). Pearson correlation coefficients (*r*) and coefficients of determination (R^2^) were determined between aphid numbers and all the phytohormone concentrations of the plants on which the aphids have fed (Appendix A). Graphpad Prism version 9.4.0 was used to perform all statistical analyses apart from the Grubbs test and correlation analysis, which were performed in XLSTAT version 2021.4.1.1185.

## Figures and Tables

**Figure 1 ijms-23-12160-f001:**
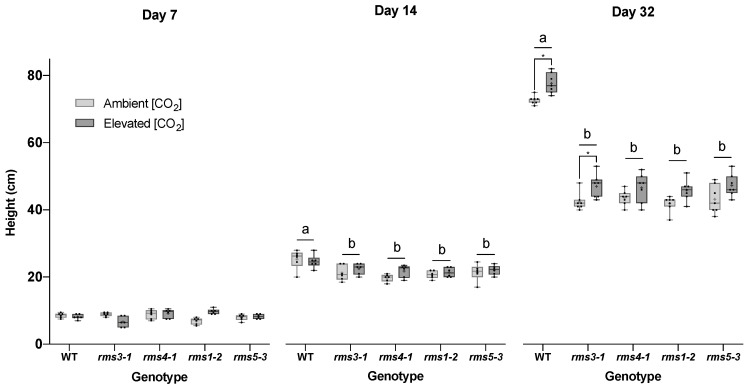
The effect of high CO_2_ on plant height in wild type peas and mutants defective either in strigolactone (SL) synthesis (*rms1*-*2* and *rms5-3*) or signaling (*rms3*-*1* and *rms4-1*). Plants were grown for up to 32 days either under ambient (420 ppm) or high CO_2_ (750 ppm). Different letters indicate a statistically significant difference (*p* < 0.05) between genotype at each time point, while a significant difference between [CO_2_] within genotype is indicated by an asterisk (*p* < 0.05). Day 7 and day 14, *n* = 6; day 32, *n* = 7. Mean indicated as +.

**Figure 2 ijms-23-12160-f002:**
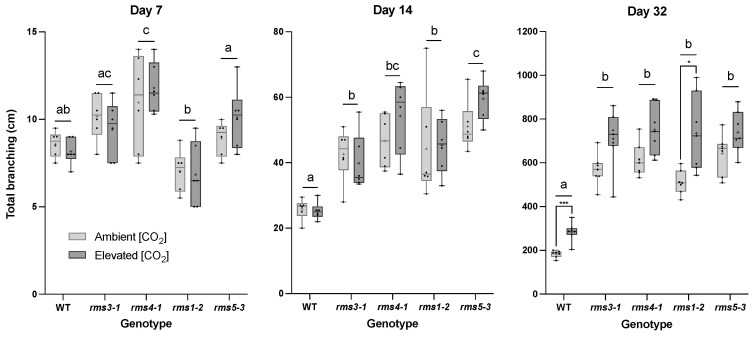
The effect of high CO_2_ on the branching of the shoots of wild type peas and mutants defective either in strigolactone (SL) synthesis (*rms1*-*2* and *rms5-3*) or signaling (*rms3*-*1* and *rms4-1*). Plants were grown for up to 32 days either under ambient (420 ppm) or high CO_2_ (750 ppm). Different letters indicate a statistically significant difference (*p* ≤ 0.05) between genotype at each time point, while a significant difference between [CO_2_] within genotype is indicated as: * *p* < 0.05; *** *p* < 0.001. Day 7 and day 14, *n* = 6; day 32, *n* = 7. Mean indicated as +.

**Figure 3 ijms-23-12160-f003:**
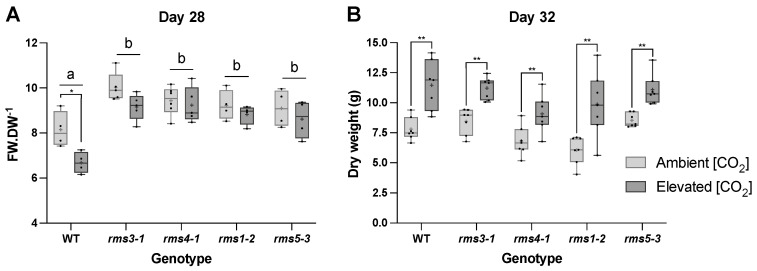
The effect of high CO_2_ on the fresh weight/dry weight ratios (**A**) and dry weights (**B**) of wild type peas and mutants defective either in strigolactone (SL) synthesis (*rms1*-*2* and *rms5-3*) or signaling (*rms3*-*1* and *rms4-1*). Plants were grown for 28 days (fresh weight/dry weight) or 32 days (dry weights) either under ambient (420 ppm CO_2_) or high CO_2_ (750 ppm). Different letters indicate a statistically significant difference (*p* < 0.05) between genotype at each time point, while a significant difference between [CO_2_] within genotype is indicated as: * *p* < 0.05; ** *p* < 0.01 (*n* = 5). Mean indicated as +.

**Figure 4 ijms-23-12160-f004:**
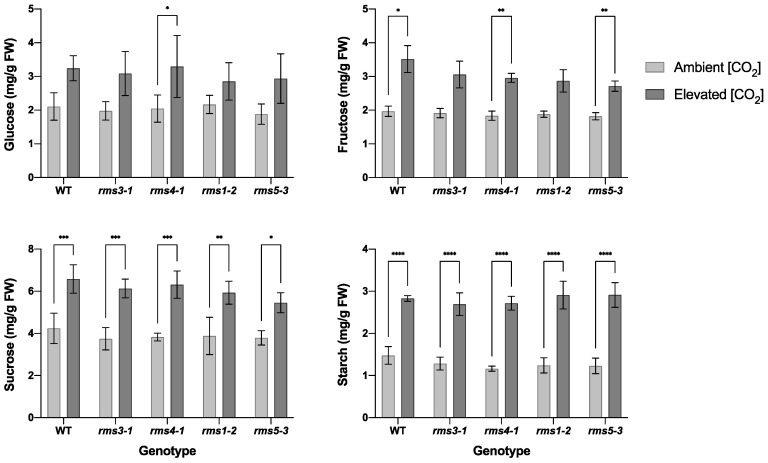
The effect of high CO_2_ on the levels of glucose, fructose, sucrose and starch in wild type peas and mutants defective either in strigolactone (SL) synthesis (*rms1*-*2* and *rms5-3*) or signaling (*rms3*-*1* and *rms4-1*). Plants were grown for 28 days either under ambient (420 ppm CO_2_; bottom row) or high (750 ppm CO_2_; top row). Data shown as mean ± SD of three replicates. Significant differences between [CO_2_] within genotype are indicated as: * *p* <0.05; ** *p* < 0.01; *** *p* < 0.001; **** *p* < 0.0001.

**Figure 5 ijms-23-12160-f005:**
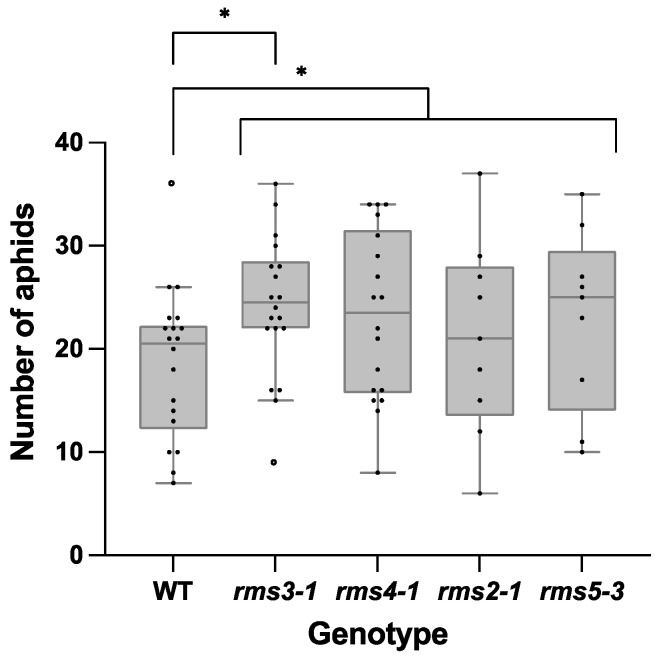
Aphid fecundity on wild type peas and mutants defective either in strigolactone (SL) synthesis (*rms1*-*2* and *rms5-3*) or signaling (*rms3*-*1* and *rms4-1*) grown under ambient CO_2_ conditions. A single pea aphid nymph was placed on each 5-day-old plant. Plants were then grown in air for 15 days before aphid numbers were counted. Significant differences between genotypes are indicated as: * *p* < 0.05. Strigolactone mutants as a group were also significantly different from the wildtype. Wildtype (WT) and *rms3-1*, *n* = 19; *rms4-1, n* = 18; *rms2-1* and *rms5-3*, *n* = 9.

**Figure 6 ijms-23-12160-f006:**
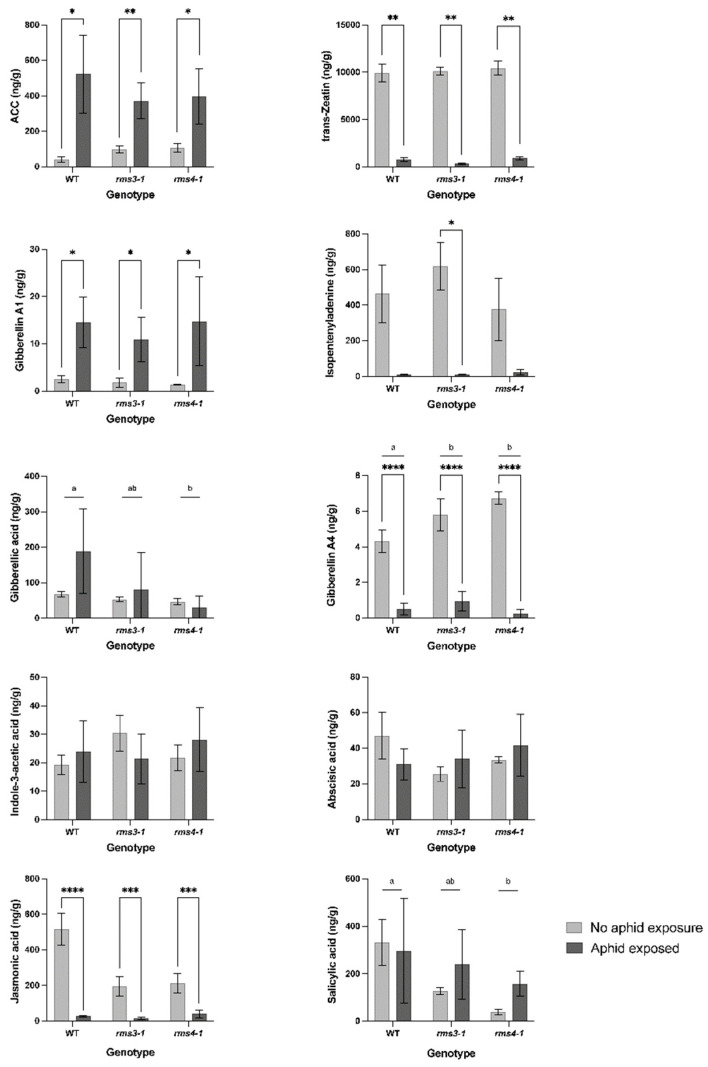
The effect aphid infestation on the levels of different phytohormones in wild type peas and mutants defective in strigolactone signaling (*rms3*-*1* and *rms4-1*) under high CO_2_ growth conditions. Data shown as mean ± SD (*n* = 3). Different letters indicate a statistically significant difference (*p* < 0.05) between genotype means (aphid-exposed and non-exposed grouped), while a significant difference between aphid exposure groups within genotype is indicated as: * *p* < 0.05; ** *p* < 0.01; *** *p* < 0.001; **** *p* < 0.0001.

**Figure 7 ijms-23-12160-f007:**
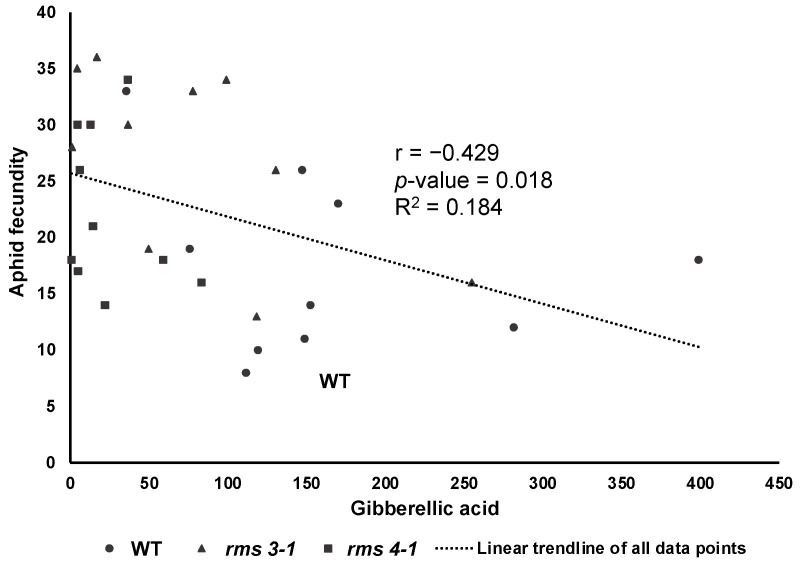
The number of aphids on each plant was significantly correlated (*p* = 0.018) with shoot gibberellic acid concentration (ng/g DW) with a single relationship explaining variation across all genotypes (*n* = 30, *n* = 10 per genotype).

## Data Availability

All data are available upon request to the authors.

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
