# Peer review of "The Effects of High CO2 and Strigolactones on Shoot Branching and Aphid–Plant Compatibility Control in Pea"

_ijms, 2022, doi:10.3390/ijms232012160_

Round 1
Reviewer 1 Report
The manuscript from Swiegers et al. describes the effect of elevated CO2 on shoot branching in pea and its interaction with hormone levels, using strigolactone synthesis or signaling mutants. The introduction is very clear, and distinctly describes the objectives of the paper. The main conclusions are (i) SL mutants are more susceptible to aphid infestation than the WT, (ii) SL deficiency impairs the accumulation or the distribution of other phytohormones (JA, SA and GA, all important for defence), and (iii) SL are not required for the shoot response to eCO2. The data and the results that are presented support these conclusions, and allow to answer to the questions previously mentioned.
I believe this manuscript should be considered for publication. I only have minor points that might be addressed before publication.
- On Figure 3, left and right panels do not correspond to the same trait, in opposition to the two previous figures. Here, it might be useful to use labels to differentiate them.
- In opposition to Figure 5, the number of replicates used to generate the results presented in Figs. 1, 2, 3, 4 and 6 is not mentioned in the legends. It should be done, to go with the rigour observed throughout the manuscript.
- The Figure 6 presents data on SL signalling mutants, in opposition to the previous figures. However, it is not justified nor mentioned why in the text.
- Statistics appearing on the graphs of the Figure 6 are confusing. For example, the statistics corresponding to the “control-aphid exposed” for the ACC assay do not match with the Table S5 (they are actually absent) ; statistics corresponding to the “control-aphid exposed” for the GA4 assay correspond to ANOVA results and do not match with the Table S5. All statistics presented on these graphs should be carefully re-annotated in agreement with the Supplementary tables.
- Concerning the data used to generate the Figure 7, it is not clearly mentioned that only the “eCO2” data have been used. It should be mentioned and perhaps discussed in the discussion, as it is not clear whether the same results would have been obtained under air condition.
- My opinion is that the main conclusions of the manuscript are as much about the effect of SL deficiency on aphid-plant relations and hormone content as about the effect of eCO2. The title of the manuscript might mention both (“The effect of high CO2 *and SL-deficient mutants* on …”).
Reviewer 2 Report
The manuscript ”The effects of high CO2 on shoot branching and aphid-plant compatibility control in pea” is dealing with mechanisms that regulate pea architecture and susceptibility to aphids under eve laved CO2. In manuscript it is confirmed that aphids interact with host plants at the molecular level and successfully suppress plant defenses. Authors compared responses of strigolactone synthesis and signaling mutants with wild type peas. They find that strigolactone are not required for shoot acclimation to eCO2 and gibberellins may be important in the defense of pea plants against aphid infestation.
Line 151: CO2conditions ...correct space
Author Response
The manuscript ”The effects of high CO2 on shoot branching and aphid-plant compatibility control in pea” is dealing with mechanisms that regulate pea architecture and susceptibility to aphids under eve laved CO2. In manuscript it is confirmed that aphids interact with host plants at the molecular level and successfully suppress plant defenses. Authors compared responses of strigolactone synthesis and signaling mutants with wild type peas. They find that strigolactone are not required for shoot acclimation to eCO2 and gibberellins may be important in the defense of pea plants against aphid infestation.
Authors: We thanks the reviewer for these positive comennts
Line 151: CO2conditions ...correct space
Authors: Amended as requested